# In Silico Identification of Novel Aromatic Compounds as Potential HIV-1 Entry Inhibitors Mimicking Cellular Receptor CD4

**DOI:** 10.3390/v11080746

**Published:** 2019-08-13

**Authors:** Alexander M. Andrianov, Grigory I. Nikolaev, Yuri V. Kornoushenko, Wei Xu, Shibo Jiang, Alexander V. Tuzikov

**Affiliations:** 1Institute of Bioorganic Chemistry, National Academy of Sciences of Belarus, 220141 Minsk, Belarus; 2United Institute of Informatics Problems, National Academy of Sciences of Belarus, 220012 Minsk, Belarus; 3Key Laboratory of Medical Molecular Virology (MOE/NHC/CAMS), School of Basic Medical Sciences, Fudan University, 131 Dong An Road, Fuxing Building, Shanghai 200032, China

**Keywords:** HIV-1 gp120 protein, cellular receptor CD4, CD4-mimetics, virtual screening, in silico click chemistry, molecular docking, quantum chemical calculations, molecular dynamics simulations, binding free energy calculations, anti-HIV-1 drugs

## Abstract

Despite recent progress in the development of novel potent HIV-1 entry/fusion inhibitors, there are currently no licensed antiviral drugs based on inhibiting the critical interactions of the HIV-1 envelope gp120 protein with cellular receptor CD4. In this connection, studies on the design of new small-molecule compounds able to block the gp120-CD4 binding are still of great value. In this work, in silico design of drug-like compounds containing the moieties that make the ligand active towards gp120 was performed within the concept of click chemistry. Complexes of the designed molecules bound to gp120 were then generated by molecular docking and optimized using semiempirical quantum chemical method PM7. Finally, the binding affinity analysis of these ligand/gp120 complexes was performed by molecular dynamic simulations and binding free energy calculations. As a result, five top-ranking compounds that mimic the key interactions of CD4 with gp120 and show the high binding affinity were identified as the most promising CD4-mimemic candidates. Taken together, the data obtained suggest that these compounds may serve as promising scaffolds for the development of novel, highly potent and broad anti-HIV-1 therapeutics.

## 1. Introduction

Human immunodeficiency virus type 1 (HIV-1) that was first identified in 1983 is the direct cause of the development of acquired immunodeficiency syndrome (AIDS) [1]. As of July 2018, the number of HIV-infected patients in the world was approximately 37 million people, with the majority of HIV infections in Asia, Africa and South America [2]. The higher incidence and prevalence of HIV infection in these countries does not reduce the relevance of the problem of HIV/AIDS for the states of North America and Europe. Although as of 2015, the pace of the development of the HIV pandemic in the world has declined, this problem still requires an urgent solution [2].

To date, more than 25 drugs have been approved for clinical use by the USA food and drug administration [3]. Depending on the mechanism of action, these drugs are divided into classes including reverse transcriptase inhibitors, proteases, integrases and entry/fusion inhibitors [3,4,5,6,7,8]. However, the extensive genetic variability in the HIV-1 envelope (Env) gene leads to the development of resistance to a particular drug some time after the start of its use [9]. This genetic diversity in HIV-1 patients is due to the high rate of viral replication, the high viral load, and the errors made in a single cycle of viral replication because of the mutations in the HIV-1 reverse transcriptase [10]. Since 1996, highly active antiretroviral therapy (HAART) has been widely used to treat HIV-1 infection [11,12]. The main goal of HAART is to overcome the resistance of the virus to individual antiretroviral drugs based on a combination of highly active therapeutics with different mechanisms of action [11,12]. Currently, HAART forms the principal methodology for treating patients with the HIV-1 infection. The use of HAART significantly increased the life expectancy of the HIV-infected patients and improved its quality, reduced the number of deaths, decreased the incidence of AIDS and HIV-related conditions [11,12]. However, the standard HAART regimens have a number of serious disadvantages, such as the toxicity of the drugs used often causing severe short-and long-term side effects (up to individual intolerance), the emergence and transmission of resistant strains, drug-drug interactions and their high cost [11,12].

The need for daily lifetime uses of several therapeutic drugs and the associated toxicity and the emergence of resistance require the development of novel, potent and effective anti-HIV agents. Most of the drugs used in HAART target the HIV-1 reverse transcriptase and protease [3,4,5,6,7,8], but these viral enzymes cannot prevent the virus from entering a target cell. This increases attention to small-molecule compounds able to inhibit the initial stages of the HIV-1 infection cycle by blocking the viral adsorption to CD4^+^ cells or/and the virus-cell membrane fusion [3,4,5,6,7,8,13,14]. The advantages of these compounds are that they create obstacles to the virus entry into new target cells, decrease the number of latent HIV-1 reservoirs and slow down the rate of the HIV-1 entry into the host cell, making the virus more sensitive to other inhibitors [3,4,5,6,7,8].

HIV-1 binds to a target cell by specific interactions of the viral Env gp120 protein with cellular receptor CD4, resulting in the conformational changes of the third variable loop of gp120 that promote the HIV-1 attachment to the chemokine co-receptors CCR5 or CXCR4 [15]. These sequential interactions of gp120 with the two host surface proteins trigger the structural rearrangements of the gp41 ectodomain which activate the Env-mediated membrane fusion [15]. From the crystal structure analysis [16], the interactions of the amino-acid residues Arg-59_CD4_ and Phe-43_CD4_ with the highly conserved residues Asp-368_gp120_, Glu-370_gp120_ and Trp-427_gp120_ are critical for the HIV-1 binding to CD4. As follows from the X-ray gp120/CD4 complex [16], Arg-59_CD4_ forms two hydrogen bonds with Asp-368_gp120_, and the Phe-43_CD4_ residue penetrates the hydrophobic pocket of the gp120 CD4-binding site, named the Phe-43 cavity, and interacts with the gp120 residues Asp-368, Glu-370, Ile-371, Asn-425, Met-426, Trp-427, and Gly-473. Importantly, the interactions of Arg-59_CD4_ and Phe-43_CD4_ with the above gp120 residues account for 23% of the total number of the HIV-1 contacts with the CD4 receptor, providing strong binding of the virus to CD4 [16].

Only two HIV-1 entry/fusion inhibitors are currently used in HAART [3,4,5,6,7,8]. These are enfuvirtide (also known as T-20) blocking the Env-mediated virus-cell membrane fusion [17] and maraviroc competing with HIV-1 for the binding to the chemokine co-receptor CCR5 [18]. However, these drugs have several disadvantages that significantly limit their use in the antiretroviral therapy [11,12], which requires the development of new, more effective and safer anti-HIV agents. To date, a large number of small-molecule compounds that inhibit the key interactions of the CD4-binding site of gp120 with CD4 have been developed [13,19,20,21,22,23,24,25,26,27,28,29,30,31,32,33,34,35,36,37,38,39]. Unfortunately, most of these compounds have not survived clinical studies for various reasons, such as the emergence of virus resistance to these drug candidates, their toxicity and high cost [13,37]. In particular, two promising HIV-1 entry inhibitors NBD-556 and NBD-557 that target the gp120/CD4 interface were discovered in 2005 using virtual screening of small molecules from commercial libraries of chemical compounds [19]. NBD-556 and NBD-557 were shown to nicely mimic most of the critical interactions of CD4 with gp120, which resulted in the design of their analogs exhibiting the higher antiviral activity and improved pharmacokinetic properties [20,21,22,23,24,25,26,27,28,29,30,31,32]. Nevertheless, the X-ray analysis of NBD-556 and its analogs bound to gp120 showed [40,41] that these small-molecule compounds target the hydrophobic Phe-43 cavity of the gp120 CD4-binding site, but cannot block the critically important H-bond/salt bridge interaction of Arg-59_CD4_ with Asp-368_gp120_ as was observed in the crystal gp120/CD4 complex [16]. Unfortunately, NBD-556 and its analogs were found to function as agonists of CD4, enhancing the infectivity of HIV-1 in CD4–CCR5^+^ cells. Despite these disappointing results, design of the HIV-1 entry inhibitors (+)-DMJ-I-228 and (+)-DMJ-II-121 indicated the possibility of developing full functional antagonists of the viral entry that target CD4-binding site of gp120 [33]. This assumption was confirmed in subsequent studies in which the HIV-1 inhibitors NBD-11021 [35], NBD-14010 [38], and NBD-14189 [39] presenting a new generation of the viral entry antagonists were developed.

Thus, despite significant progress in the development of novel potent HIV-1 entry/fusion inhibitors, there are currently no licensed antiviral drugs based on inhibiting the critical interactions of the HIV-1 envelope gp120 protein with cellular receptor CD4. In this connection, studies on the design of new small-molecule compounds able to block the binding of gp120 to CD4 are still of great value.

Over the past decades, computational modeling methods have become an important element in drug design and drug discovery [42,43,44,45,46] allowing one to significantly reduce the time and cost required for developing new therapeutics [47]. The choice of strategy for successful application of pharmacology modeling in the design of novel drugs depends on the type of initial data available which should contain information on the structure of molecular target and/or on known bioactive compounds [48,49,50]. The most popular in silico approach to the identification of new drug candidates is to search for small-molecule compounds in commercial molecular libraries by virtual screening [48,49]. This approach makes it possible to discover compounds with the required structural and pharmacophore features, but usually the values of their biological activity turn to be quite low [48,49]. Notwithstanding, these compounds may be used as good scaffolds for quantitative structure-activity relationship optimization or as modular units for design of new molecules with higher biological potency and improved pharmacokinetic profile. In doing so, the application of the concept of click chemistry allowing one to generate a large number of drug candidates by their assembly from small modular units [51,52,53,54] seems very promising. The click chemistry methodology has been implemented in silico in the AutoClickChem software package [55].

In this work, in silico design of drug-like compounds containing the moieties that make the ligand active towards gp120 was performed within the concept of click chemistry. Structural models of the designed molecules bound to gp120 were then generated by molecular docking and optimized using semiempirical quantum chemical calculations. Finally, the binding affinity analysis of these ligand/gp120 complexes was performed by molecular dynamic simulations and binding free energy calculations. As a result, five top-ranking compounds that showed strong attachment to the CD4-binding site of gp120 by mimicking the key interactions of CD4 with gp120 were identified as the most promising CD4-mimemic candidates.

## 2. Materials and Methods

### 2.1. In Silico Design of Small-Molecule CD4-Mimetic Candidates

In silico design of potential CD4-mimetics was carried out by the AutoClickChem program [55]. In the first stage of the study, virtual screening of the “Drug-Like subset” of the ZINC database [56] was performed to form two molecular libraries of small modular units with the functional groups involved in the reaction of azide-alkyne cycloaddition (Figure 1). To do this, the DataWarrior program [57] providing data visualization and analysis was employed. Using DataWarrior, small aromatic molecules with molecular mass <250 Da containing azide or alkyne groups were selected from ZINC and placed in library 1 (Figure 1). In the same way, all low-molecular compounds (molecular mass <250 Da) with the above functional groups were collected in library 2 (Figure 1). The choice of aromatic molecules as modular units for the design of CD4-mimetic candidates is due to the fact that their aromatic moieties can mimic the key interactions of Phe-43_CD4_ with the Phe-43 cavity of gp120 that dominate the HIV-1 binding to CD4. According to the X-ray data [16], the benzene ring of Phe-43_CD4_ is buried into the Phe-43 cavity of gp120, resulting in the blockade of the gp120 residues critically important for viral adsorption to CD4^+^ cells. In addition, novel HIV-1 entry antagonists, such as NBD-11021, NBD-14010 and NBD-14189, show the similar interaction modes to bind this hydrophobic pocket of gp120 [35,38,39].

As a result of screening of the ZINC database, a total of 1388 and 3769 compounds were included in libraries 1 and 2, respectively. These small modular units were then used as reactants to mimic the click-reaction of azide-alkyne cycloaddition [51] by AutoClickChem [55], resulting in a combinatorial library of 1,655,301 hybrid molecules in which 294,378 compounds fully satisfied the Lipinski’s “rule of five” [58]. These 294,378 drug-like molecules were further screened by molecular docking, quantum chemical calculations and molecular dynamics simulation to evaluate the affinity of their binding to gp120 and select the molecules most promising for synthesis and biochemical assays (Figure 1).

### 2.2. Molecular Docking

Molecular docking of the designed compounds with gp120 was performed by the QuickVina 2 program [59] in the approximation of rigid receptor and flexible ligands. The HIV-1 inhibitor NBD-11021 (Figure 2) which is the lead viral entry antagonist [35] was used in the calculations as a positive control. The 3D structure of gp120 was isolated from the crystal complex of this glycoprotein with CD4 and antibody 17b (the PDB file 1GC1; http://www.rcsb.org/pdb/) [16]. The 3D structure of NBD-11021 was taken from the X-ray complex of this compound with gp120 (the PDB file 4RZ8) [35]. The gp120 and ligand structures were prepared by adding hydrogen atoms with the OpenBabel software [60] followed by their optimization in the UFF force field [61]. The ligands were docked to the crystal gp120 structure [16] using QuickVina 2 [59]. The grid box included the Phe-43 cavity of the gp120 CD4-biding site and was the region of the crystal structure [16] with the following boundary X, Y, Z values: X ∈ (24 Ǻ; 34 Ǻ, Y ∈ (−15 Ǻ; −5 Ǻ), Z ∈ (78 Ǻ; 88 Ǻ). The value of the “exhaustiveness” parameter defining the number of individual sampling “runs” was set to 50 [59]. For each ligand, the docked structures with the best scores were analyzed to identify compounds with the values of binding free energy less than −7 kcal/mol. As a result, 100 ligand/gp120 complexes were selected for further optimization by quantum chemical calculations.

### 2.3. Quantum Chemical Studies

The quantum chemical optimization of the selected docking models was implemented by the semiempirical quantum chemical method PM7 [62] associated with the MOPAC2016 program [63]. Before the calculations, the ligand/gp120 complexes were supplemented with hydrogen atoms and optimized in the UFF force field [61]. For this purpose, the OpenBabel program [60] was used. The calculations were performed in the COSMO solvation model (COnductor-like Screening MOdel) approximation [64,65,66] in an implicit solvent with water’s dielectric constant of 78.4 [63]. To speed up the calculations, the Localized Molecular Orbitals method [67,68] available in MOPAC in the form of the linear scaling SCF MOZYME algorithm [62,63] was applied. The value of RMS gradient was set to 10 kcal/mol/Ǻ [63]. In the final step, the PM7-based complexes were characterized in terms of the ligand/gp120 interaction profile and the values of dissociation constants (K_d_) and binding energies. Based on the data obtained, five compounds that exhibited the higher binding affinity compared with NBD-11021 were selected for the MD simulations.

### 2.4. Analysis of the PM7-Based Ligand/gp120 Complexes

The binding modes of the identified compounds with gp120, namely hydrogen bonds, salt bridges, van der Waals contacts and T-stacking interactions were characterized by the BINANA program [69]. The ligand poses in the quantum-based gp120/ligand complexes were visualized with the program UCSF Chimera [70]. To visualize van der Waals contacts, the program LigPlot [71] was employed. The values of K_d_ and binding free energies for the ligand/gp120 structures were calculated using the scoring functions of NNScore 2.0 [72] and QuickVina 2 [59], respectively.

### 2.5. Molecular Dynamics Simulations

The classical dynamics of the ligand/gp120 complexes in water was made with the implementation of Amber 11 using the Amber ff10 force field [73]. The ANTECHAMBER module was employed to set the Gasteiger atomic partial charges [73]. To prepare the force field parameters, the general AMBER GAFF force field [74] was used. Hydrogen atoms were added to gp120 by the tleap program of the AMBER 11 package [73]. Initially, the ligand/gp120 complexes were each placed in an octahedron box with periodic boundary conditions. In addition to the ligand/gp120 complex, the box for the MD simulations included TIP3P water [75] as an explicit solvent, Na^+^ and Cl^−^ ions providing overall salt concentration of 0.15 M. After setting up the system, an energy minimization was performed using 500 steps of the steepest descent algorithm followed by 1000 steps of the conjugate-gradient method. The atoms of the complex assembly were then fixed by an additional harmonic potential with the force constant of 1.0 kcal/mol and the system was subject to the equilibration phase. The system equilibration was carried out in three consecutive stages: (1) the system was gradually heated from 0 K to 310 K for 1 ns in NVT ensemble using a Langevin thermostat with a collision frequency of 2.0 ps^−1^ [73]; (2) pressure equilibration was made for 1 ns at 1.0 bar in NPT ensemble using Berendsen barostat with a 2.0 ps characteristic time [73]; (3) the constraints on the complex assembly were removed and the system was equilibrated again at 310 K over 2 ns under constant volume conditions. After equilibration was achieved, the MD simulations were carried out for 30 ns in NPT ensemble at temperature T = 310 K and P = 1 bar. Bonds involving hydrogen atoms were constrained using SHAKE algorithm [76] to achieve the integration time-step of 2 fs. Long-range electrostatic interactions were calculated using Particle Mesh Ewald (PME) algorithm [77]. Coulomb interactions and van der Waals interactions were truncated at 10 Å.

### 2.6. Binding Free Energy Calculations

The values of binding free energy and contributions from each gp120 residue to its enthalpic component were calculated with AMBER 11 [73] using the MM/GBSA method [78,79,80]. Intermolecular hydrogen bonds appearing in the MD trajectories of the analyzed complexes were identified by the ptraj procedure of AMBER 11 [73]. The calculations were made for 500 snapshots extracted from the final 25 ns of the MD trajectories, by keeping the snapshots every 50 ps. The polar solvation energies were computed in continuum solvent using Poisson-Boltzmann continuum-solvation model with ionic strength of 0.1. The non-polar terms were estimated using solvent accessible surface areas [81]. The Nmode module in Amber 11 was applied to calculate the entropy term of the binding free energy [73].

## 3. Results and Discussion

Analysis of the data obtained revealed 5 top-ranking compounds exhibiting the high-affinity binding to the HIV-1 gp120 protein and mimicking the critical interactions of the virus with primary receptor CD4. These molecules were therefore selected for the final analysis as the most promising CD4-mimetic candidates (Figure 3). Physicochemical properties of the identified compounds associated with the Lipinski’s “rule of five” [58] are given in Table 1, and Figure 4 casts shed on the scheme of computer-aided assembly of these hybrid molecules. As follows from Table 1, the ADME parameters of these compounds that provide highly important characteristics for a potential drug, such as absorption, distribution, metabolism and excretion, fully satisfy the requirements of the “rule of five” [58].

Insights into the PM7-based ligand/gp120 complexes show (Figure 5) that all the selected molecules mimic the key interactions of Phe-43 and Arg-59 of CD4 with the Phe-43 cavity of gp120 and residue Asp-368_gp120_ located within the vestibule of this hydrophobic pocket [16]. It is known that these interactions greatly contribute to the attachment of HIV-1 to a target cell [16]. This is also confirmed by the data on site-directed mutagenesis [83,84], whereby single substitutions of Phe-43_CD4_ and Arg-59_CD4_ for alanine result in a 550-fold and 9-fold reduction of binding affinity to gp120, respectively.

The calculations of binding modes appearing in the ligand/gp120 complexes predict intermolecular interactions involving the gp120 residues that dominate the HIV-1 binding to CD4 (Table 2). In particular, analysis of the hydrogen-bonding network in the ligand/g120 interface indicates (Figure 5, Table 2) that the compounds of interest form hydrogen bond with Asp-368_gp120_ critically important for the successful development of viral entry antagonists [33]. This mode of interaction between Asp-368_gp120_ and Arg-59_CD4_ was found in the crystal gp120 structure in complex with CD4 and neutralizing human antibody 17b [16].

In addition, compound IV make a salt bridge with Asp-368_gp120_ which was also found between this residue of gp120 and Arg-59 of CD4 in the crystal gp120–CD4 complex, and compounds II and V participate in specific T-shaped π-π-interactions with the functionally important conserved residue Trp-427_gp120_ (Table 2) [16]. Besides the H-bond and salt bridge with Asp-368_gp120_, compound IV is involved in the hydrogen bonding with the residue Met-426_gp120_ that is an essential component of the gp120-CD4 binding hotspots [34]. It is known that the direct H-bond interaction between the ligand and the backbone carbonyl of Met-426_gp120_ leads to an improvement in the anti-HIV-1 potency compared to the dual hotspot antagonists blocking Met-426_gp120_ via a water-mediated H-bond [33].

According to the predicted binding modes, all the designed CD4-mimetic candidates participate in numerous van der Waals interactions with the gp120 residues that play a key role in the virus binding to Phe-43 of CD4 (Table 2, Figure 6). These gp120 residues are Thr-257, Asp-368, Glu-370, Asn-425, Met-426, Trp-427, Gly-473 and Met-475 (Table 2, Figure 6). One of the aromatic rings of the analyzed compounds is buried into the Phe-43 cavity (Figure 4) mimicking the pivotal interactions of the benzene group of Phe-43_CD4_ with this hydrophobic pocket of the CD4-binding site of gp120. It is important to note that the specified residues of gp120 are the dominant contributors to the gp120/CD4 interaction [16]. From the X-ray data [16], Arg-59_CD4_ is involved in the H-bonding with Asp-368_gp120_, and Phe-43_CD4_ forms a wide network of van der Waals contacts with Glu-370_gp120_, Asn-425_gp120_, Met-426_gp120_, Trp-427_gp120_, and Gly-473_gp120_. This observation also concerns the gp120 residue Thr-257 located in the crystal gp120/CD4 structure under the benzene ring of Phe-43_CD4_ as well as Met-475_gp120_ [16]. In addition to the above gp120 residues, individual compounds form the direct interatomic contacts with the other residues of gp120 which interact with CD4 as well. These residues are Ile-371 (compounds IV, V), Val-430 (compounds I, II, III, V), and Gly-431 (compounds I, III) (Table 2, Figure 6).

Thus, inspection of the intermolecular interaction profile predicted for the ligand/gp120 structures shows that the identified CD4-mimetic candidates exhibit close modes of binding to the HIV-1 envelope gp120, resulting in the blockade of the two well-conserved vulnerable spots of this glycoprotein—the Phe-43 cavity and Arg-368. This binding is similar to that of CD4 to gp120 and mainly provided by multiple van der Waals contacts with the functionally important residues of the Phe-43 cavity, hydrogen bonds with Arg-368_gp120_ and Met-426_gp120_ (compound IV), salt bridge with Arg-368_gp120_ (compound IV) and T-shaped-interactions between π-conjugated systems of compounds II, V and Trp-427_gp120_ (Table 2, Figure 5 and Figure 6). Among these binding modes, intermolecular van der Waals interactions are the major contributors to the ligand/gp120 interface, and hydrogen bond involving Asp-368_gp120_ is significant for the inhibition of viral entry without unwanted allosteric signal (Table 2, Figure 5).

The data on the intermolecular interaction network are in line with the results of binding affinity prediction obtained from the analysis of the PM7-based ligand/gp120 complexes using NNScore 2.0 and QuickVina 2 (Table 3). From the data of Table 3, the analyzed complexes show the low values of K_d_ and binding free energies, suggesting strong attachment of the designed CD4-mimetic candidates to gp120. Furthermore, these values are lower than those calculated for the NBD-11021/gp120 complex using the identical computational protocol (Table 3). However, when comparing the data of Table 3, one needs to keep in mind that all computational methods for prediction of binding affinity rely on a number of approximations making a mathematical solving the problem possible. These approximations include the use of simplified forms of the first-principles equations, limitations of the size of the system (for example, periodic boundary conditions), fundamental approximations to the underlying equations, etc. Nevertheless, the accuracy of the semiempirical quantum chemical method PM7 [62,63] gives reason to suppose that the values of K_d_ and binding free energies predicted for the designed compounds bound to gp120 (Table 3) are lower than those calculated for the NBD-11021/gp120 complex. This assumption is supported by the close values of binding free energies estimated by the classical force field of QuickVina 2 [59] and a neural-network-based scoring function NNScore 2.0 [72] (Table 3) as well as by the results of a recent study [85] according to which the use of molecular docking with classical force field in combination with the semiempirical quantum chemical method PM7 significantly improves the ligand positioning accuracy.

Molecular dynamics insights into the ligand/gp120 complexes support the principal conclusions made from the analysis of their static models. These complexes are relatively stable within the MD simulations, as evidenced by the values of the root-mean square deviations (RMSD) of the atomic positions for the dynamic and static models of the designed compounds bound to gp120 (Figure 7), as well as by the averages of binding free energies, their enthalpic terms, and corresponding standard deviations (Table 4). Analysis of Figure 7 indicates that these complexes do not undergo significant structural rearrangements on the MD trajectories: the averages of the RMSD change from 2.28 ± 0.39 Å (compound III) to 3.07 ± 0.60 Å (compound IV), testifying to their relative conformational stability. Given the MM/GBSA method errors [78,79,80], one can suggest that the dynamic structures of compounds III–V in the complexes with gp120 show the averages of binding free energy similar to the value calculated for the HIV-1 inhibitor NBD-11021 (Table 4). Furthermore, these averages are close to the values predicted by QuickVina 2 [59] and NNScore 2.0 [72] for the static structures of compounds III–V bound to gp120 (Table 3 and Table 4) as well as to the value of −9.5 ± 0.1 kcal/mol measured for the CD4/gp120 complex by isothermal titration calorimetry [87]. According to the MM/GBSA calculations, ligand II exhibits the higher binding affinity compared with compounds III–V, and ligand I demonstrates much lower average of binding free energy than the other identified compounds and NBD-11021 (Table 4). In this regard, it should be noted that the QuickVina/PM7 calculations also identify compound I as the most potent CD4-mimetic candidate exposing the lowest values of K_d_ and binding free energy among the analyzed compounds (Table 3).

Analysis of the data on the contributions of individual gp120 amino acids into the enthalpic term of binding free energy reveals the residues of gp120 dominating the ligand/gp120 interaction profile. Table 5 shows that these residues are Glu-370, Asn-425, Met-426, Trp-427, Gly-473, Asp-474, and Met-475, in agreement with the findings of the QuickVina/PM7 calculations (Table 2, Figure 5 and Figure 6).

Importantly, it is those residues that greatly contribute to the CD4/gp120 interaction [16]. In favor of the results obtained is also evidence of the data on the hydrogen-bonding network in the dynamic ligand/gp120 structures demonstrating the high percentage occupancies of intermolecular H-bonds involving the residues of gp120 that play a pivotal role in the HIV-1 attachment to CD4 (Table 6).

It should be noted that the data presented above were obtained using the X-ray CD4/gp120/Fab17b structure, as a target for prediction of the interaction modes and binding affinity profiles of the designed compounds and gp120. This gp120 core structure that was determined in 1998 [16] provided a foundation for rational design and synthesis of the HIV-1 entry inhibitors targeting CD4-binding site of gp120. However, most of the initial gp120-directed inhibitors were developed employing the CD4-bound conformation of gp120 presenting a form of gp120 in which the V1, V2, and V3 loops and N and C termini have been truncated [40]. Determination of an extended, unliganded gp120 core structure [40] indicating the similarity between the CD4-bound and unliganded structures of the gp120 core partially resolved this dilemma. This observation was later confirmed by comparative analysis of various cryo-ET and single-particle cryo-EM trimer constructs that revealed the overall architecture of the Env trimer in both the unliganded and bound states [88,89,90]. The computational evaluation of different HIV-1 gp120 conformations as targets for de novo docking of first-and second-generation small-molecule CD4 mimics showed [91] that in silico neutralization of the highly conserved residue Asp-368 of gp120 is critically necessary to obtain the correct orientation of small-molecule CD4 mimetics in their binding site when docking against the monomeric gp120 core, which correlates with IC50’s measured in CD4 binding competition ELISA and with KD’s measured on gp120 core monomer. As shown above, all the identified compounds form hydrogen bond and van der Waals contacts with Asp-368_gp120_, suggesting that the data obtained from the analysis of the docked ligand/monomeric gp120 complexes provide correct information on the interaction modes and binding affinity profiles of these molecules and gp120. This assumption is supported by the optimization of these complexes using the semiempirical quantum chemical method PM7 which improves the accuracy of docking models based on classical force field, as evidence from the study of Sulimov & coauthors [86]. Furthermore, the findings of molecular dynamics of the analyzed docked structures that represent the flexibility of both ligands and a target protein also validate the assumption made. Finally, there are some active anti-HIV compounds that were obtained by virtual screening using the X-ray crystal structure of the CD4-bound HIV-1 gp120 core and have provided lead scaffolds for successful development of novel potent HIV-1 entry inhibitors (e.g., [33,92], indicating the validity of application of this structure to screen new CD4 mimetics candidates targeting CD4-binding site of gp120. In particular, docking studies [23] using the X-ray structure of the CD4/gp120_core_/Fab17b complex [16] showed that the agonist of the viral entry NBD-556 [19] binds to the Phe-43 cavity and the para-chlorophenyl ring of this compound penetrates the cavity more deeply compared with the benzene ring of the Phe-43_CD4_ in the CD4/gp120core/Fab17b structure. These results led to the assumption that congeners of NBD-556 with enhanced affinity for gp120 might compete more efficiently with CD4 for binding to gp120 and prevent the downstream allosteric events in the entry cascade [33], giving impetus to the development of a new class of the NBD-based functional antagonists of the viral entry that target CD4-binding site of gp120 [35,38,39].

## 4. Conclusions

The data of molecular docking combined with the quantum chemical calculations and molecular dynamics studies indicate that the designed compounds (Figure 3) exhibit the similar modes of binding to the HIV-1 envelope gp120 (Table 2), resulting in destruction of the critical interactions of Phe-43_CD4_ and Arg-59_CD4_ with the two well-conserved hotspots of the CD4-binding site of gp120, namely the Phe-43 cavity and Asp-368_gp120_. These binding modes are mainly provided by numerous van der Waals interactions with the gp120 residues Thr-257, Asp-368, Glu-370, Asn-425, Met-426, Trp-427, Gly-473 and Met-475, and the hydrogen bond with Asp-368_gp120_ that is associated with increasing the binding affinity without triggering the undesirable allosteric signal [33] is also highly important (Table 2, Figure 5 and Figure 6). The selected CD4-mimetic candidates fully satisfy the Lipinski’s “rule of five” (Table 1) and show strong attachment to the CD4-binding site of gp120, in line with the data on the predictions of binding affinity in terms of K_d_ and binding free energy (Table 3 and Table 4). Among the designed CD4-mimetic candidates, it should be emphasized compound I (Figure 3) that exposes the lowest value of binding free energy compared with the other identified molecules and NBD-11021 (Table 4). This compound also exhibits the best value of K_d_ (Table 3) and the widest hydrogen-bonding network appearing in the MD trajectory of the ligand/gp120 complexes (Table 6). This small-molecule hit is therefore a higher-priority candidate for detailed experimental evaluation.

Obviously, the above computational data are a prediction and their final confirmation can be obtained only after testing the designed molecules for anti-HIV activity. Unfortunately, synthetic methodologies still limit the compounds that computational chemists can design. However, five compounds identified in this work can be synthesized by the click-reaction of azide-alkyne cycloaddition using commercially available modular units as reagents (Figure 4). This study is now in progress and its further advancement suggests to use these CD4 mimetic candidates as the fixed scaffolds for computer-based generation of their analogs with improved antiviral activity and pharmacokinetic profile followed by synthesis and detailed biochemical assays. At the same time, the lead compound optimization is assumed to perform using the 4NCO trimeric gp120 structure [93] that is more appropriate target for structure-based design of the gp120-directed inhibitors compared with the 4TVP trimer and gp120 core monomer [91].

Taken together, the data obtained suggest that the identified compounds (Figure 4) may serve as promising scaffolds for the development of novel, highly potent and broad anti-HIV-1 therapeutics.

## Figures and Tables

**Figure 1 viruses-11-00746-f001:**
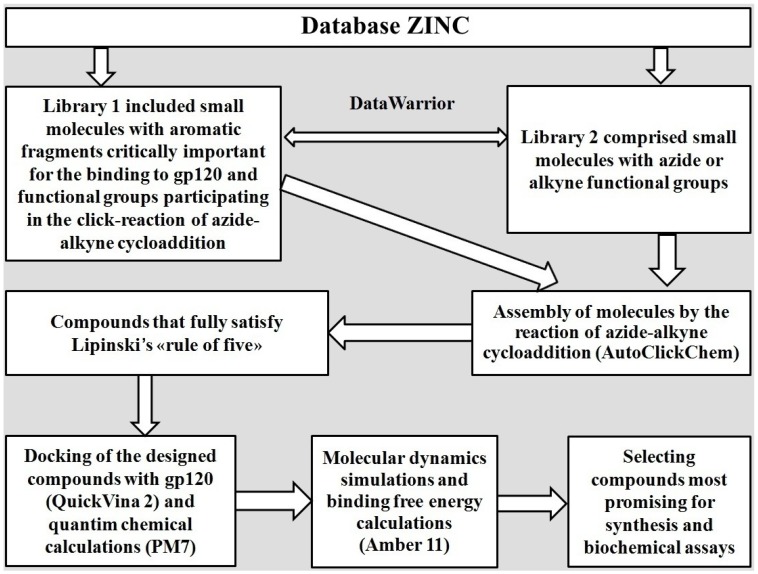
The algorithm scheme used for the identification of CD4-mimetic candidates.

**Figure 2 viruses-11-00746-f002:**
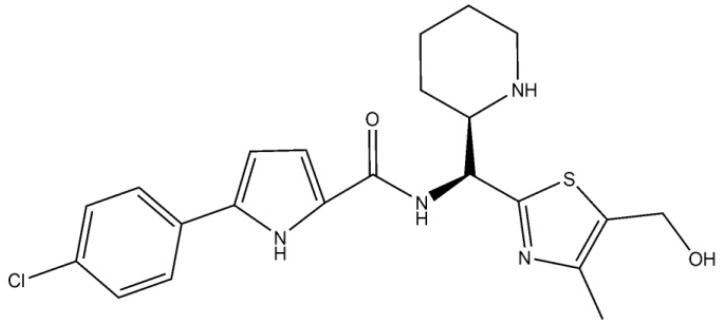
Chemical structure of NBD-11021, the lead viral entry antagonist that blocks gp120–CD4 interaction with pan-neutralization of diverse subtypes of clinical isolates (IC_50_ as low as 270 nM) [35].

**Figure 3 viruses-11-00746-f003:**
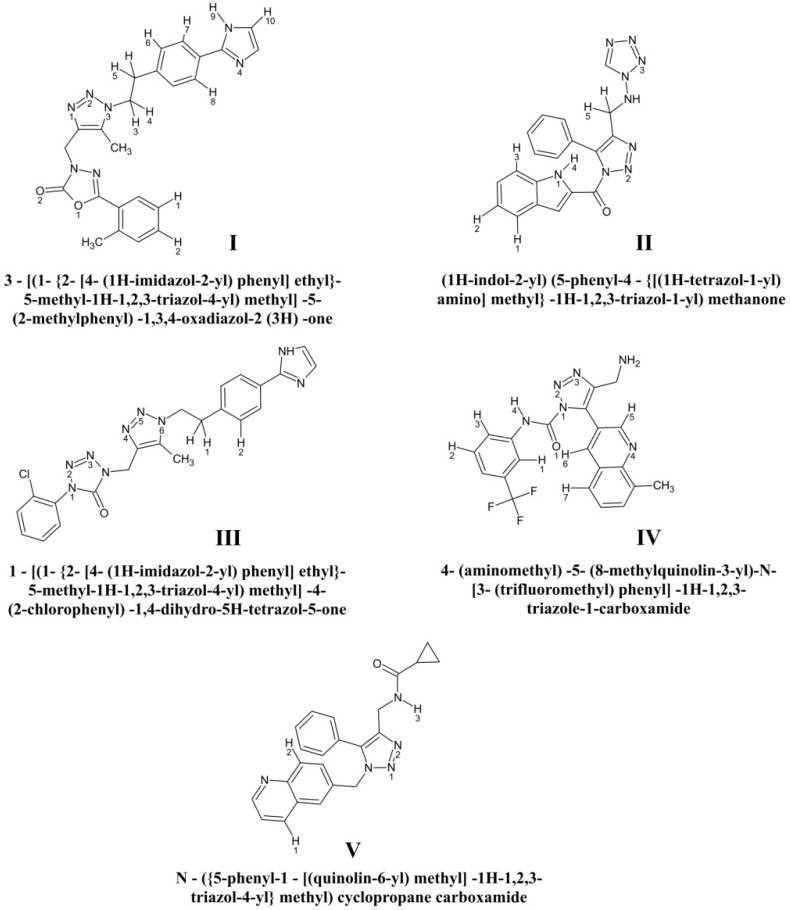
Chemical structures of the most promising CD4-mimetic candidates. Systematic names of these compounds are given. The hydrogen, oxygen and nitrogen atoms involved in the hydrogen bonds in the dynamic ligand/gp120 complexes are numbered (see the text).

**Figure 4 viruses-11-00746-f004:**
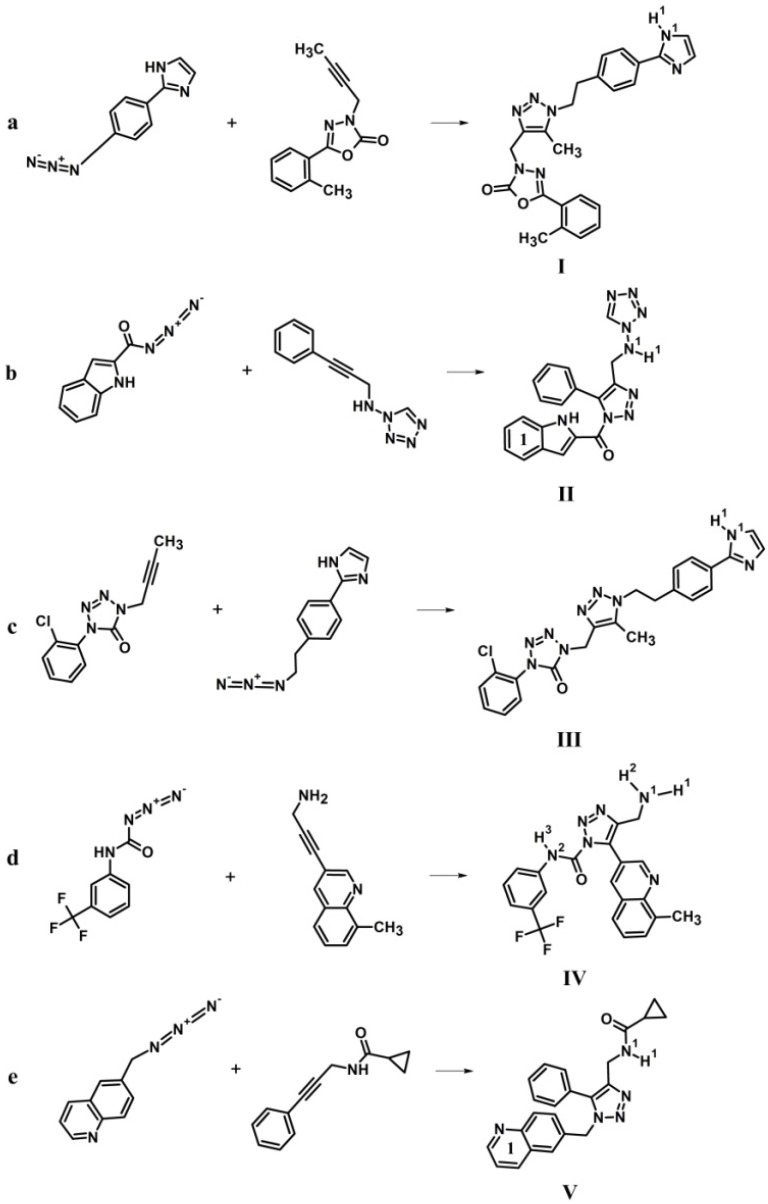
Computer-aided assembly of hybrid molecules I (**a**), II (**b**), III (**c**), IV (**d**), V (**e**) that were identified as the most promising CD4-mimetic candidates. The reagents and products of the reaction of azide-alkyne cycloaddition are shown. Functional groups of the designed molecules forming hydrogen bonds and salt bridges with gp120 are indicated using superscripts (Table 2). The aromatic rings involved in the T-shaped-interactions with Trp-427_gp120_ are numbered (Table 2).

**Figure 5 viruses-11-00746-f005:**
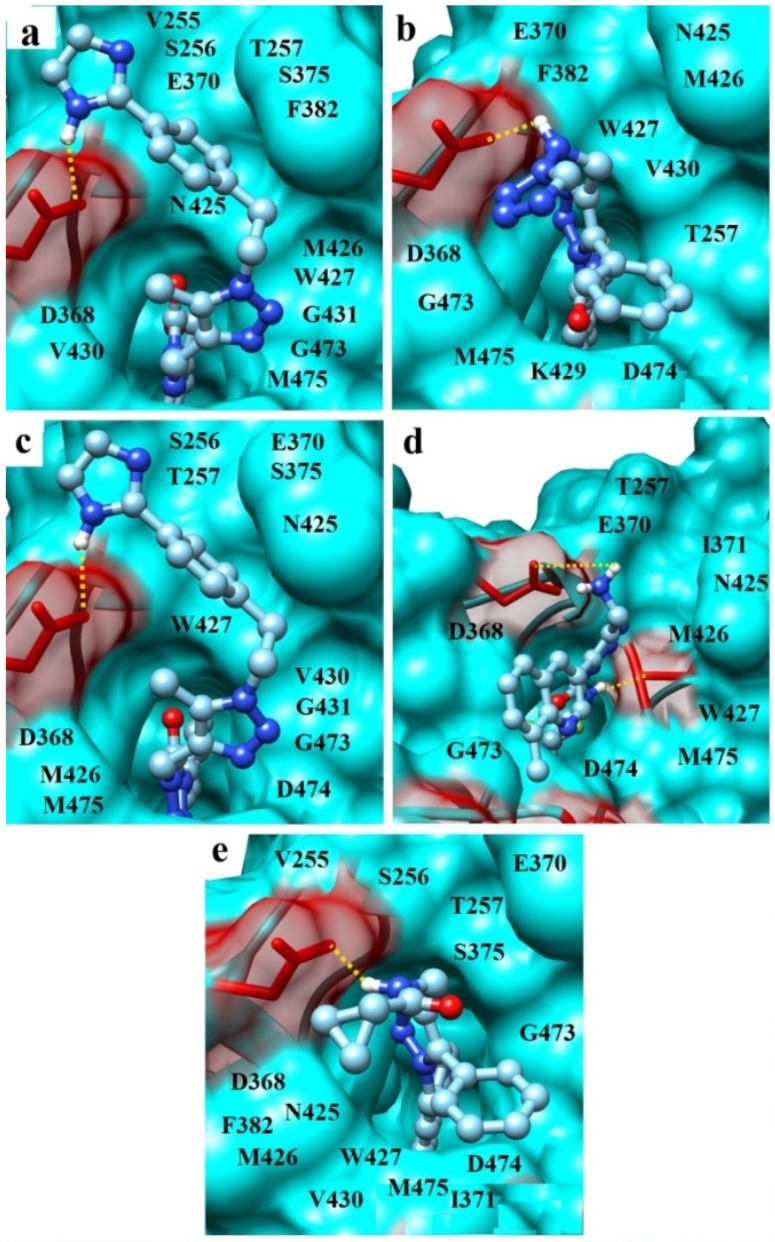
The PM7-based complexes of compounds I (**a**), II (**b**), III (**c**), IV (**d**) and V (**e**) with gp120. The Phe-43 cavity of gp120 and the residues located within the vestibule of this hydrophobic pocket are shown. The residues of gp120 forming hydrogen bonds and van der Waals contacts with the CD4-mimetic candidates are indicated (see Table 2, Figure 6). Hydrogen bonds are marked by dotted lines.

**Figure 6 viruses-11-00746-f006:**
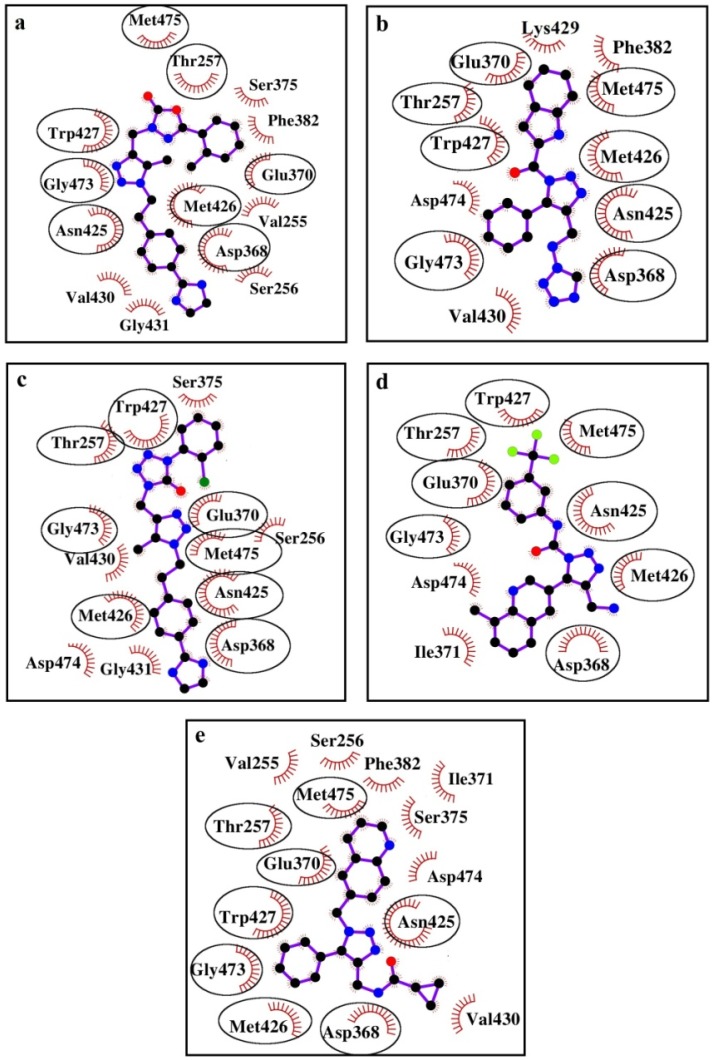
The gp120 residues making van der Waals contacts with compounds I (**a**), II (**b**), III (**c**), IV (**d**), and V (**e**). Residues involved in van der Waals interactions in all of the cases of interest are marked by circles. The total number of van der Waals contacts is: 80 (**a**), 66 (**b**), 70 (**c**), 61 (**d**), and 80 (**e**).

**Figure 7 viruses-11-00746-f007:**
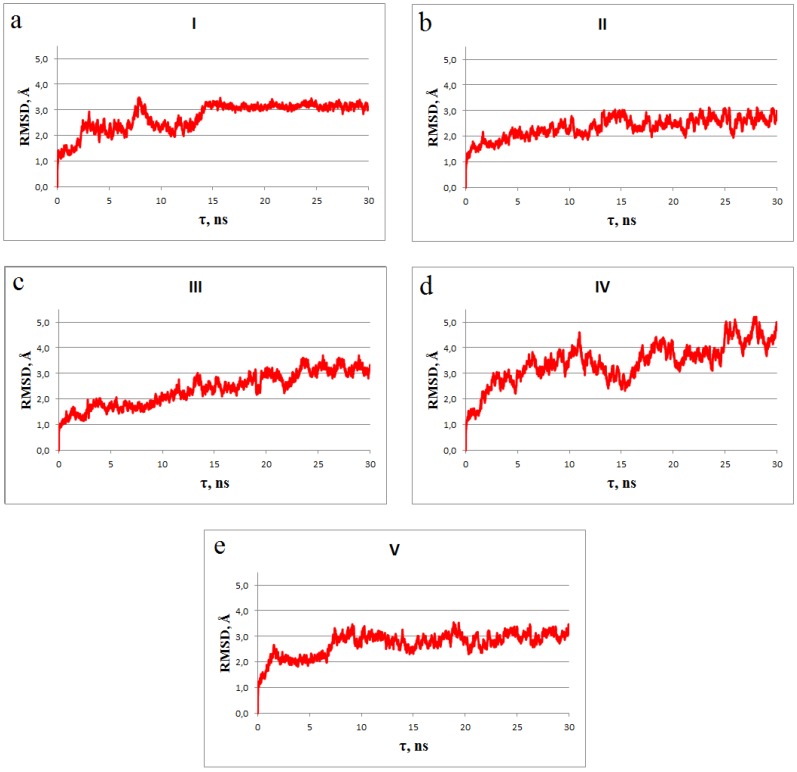
The time dependence of the root-mean square deviations (RMSD) (Å) computed between all of the MD structures and the stating models of the ligand–gp120 complexes. The backbone atoms of gp120 were used in the calculations. The RMSD averages are: compound I–gp120 complex (**a**) −2.71 ± 0.31 Å, compound II–gp120 complex (**b**) −2.30 ± 0.16 Å, compound III–gp120 complex (**c**) −2.28 ± 0.39 Å, compound IV–gp120 complex (**d**) −3.07 ± 0.60 Å, and compound V–gp120 complex (**e**) −2.67 ± 0.21 Å.

**Table 1 viruses-11-00746-t001:** CD4-mimetic candidates and their physicochemical parameters associated with the Lipinski’s “rule of five” ^1^.

Compound	Systematic Name	Chemical Formula	Molecular Mass (Da)	Log P ^2^	Number of H-Bond Donors	Number of H-Bond Acceptors
I	3-[(1-{2-[4-(1*H*-imidazol-2-yl) phenyl]ethyl}-5-methyl-1*H*-1,2,3-triazol-4-yl) methyl]-5-(2-methylphenyl)-1,3,4-oxadiazol-2 (3*H*)-one	C_24_H_23_N_7_O_2_	419.02	2.634	1	9
II	(1*H*-indol-2-yl) (5-phenyl-4-{[(1*H*-tetrazol-1-yl)amino]methyl}-1*H*-1,2,3-triazol-1-yl) methanone	C_19_H_15_N_9_O	372.04	1.131	1	10
III	1-[(1-{2-[4-(1*H*-imidazol-2-yl) phenyl] ethyl}-5-methyl-1*H*-1,2,3-triazol-4-yl) methyl]-4-(2-chlorophenyl)-1,4-dihydro-5*H*-tetrazol-5-one	C_22_H_20_ClN_9_O	419.03	0.125	1	8
IV	4-(aminomethyl)-5-(8-methylquinolin-3-yl)-*N*-[3-(trifluoromethyl)phenyl]-1*H*-1,2,3-triazole-1-carboxamide	C_21_H_17_F_3_N_6_	412.03	0.905	2	7
V	*N*-({5-phenyl-1-[(quinolin-6-yl) methyl]-1*H*-1,2,3-triazol-4-yl}methyl)cyclopropane carboxamide	C_23_H_21_N_5_O	363.02	1.120	1	6

Notes: ^1^ The data given were obtained using an open source virtual screening tool DruLiTo [82]; ^2^ The compound lipophilicity.

**Table 2 viruses-11-00746-t002:** Intermolecular interactions in the optimized models of the CD4-mimetic candidates bound to the gp120 protein.

Compound	Hydrogen Bond ^1^	Van der Waals Contacts ^2^	Salt Bridges and π-π-Interactions ^3^
I	N^1^H^1^ ...O_D2_[D_368_]	V255(5), S256(2), T257(3), D368(5), E370(8), S375(5), F382(2), N425(11), M426(5), W427(12), V430(7), G431(3), G473(4), M475(8)	-
II	N^1^H^1^ ...O_D1_[D_368_]	D368(4), E370(7), F382(2), N425(8), M426(5), W427(7), V430(5), G473(7), M475(5), D474(7), K429(5), T257(4)	1...W427 (T-shaped-interaction)
III	N^1^H^1^ ...O_D2_[D_368_]	S256(2), T257(3), D368(5), E370(8), S375(2), N425(7), M426(8), W427(13), V430(5), G431(3), G473(4), M475(2), D474(8)	-
IV	N_1_H_1_...O_D2_[D_368_] N^2^H^3^...O[M_426_]	T257(2), D368(6), E370(8), I371(7), N425(6), M426(4), W427(9), M475(6), G473(8), D474(5)	N^1^H^1^H^2^...D368 (salt bridge)
V	N^1^H^1^ ...O_D2_[D_368_]	V255(3), S256(2), T257(5), D368(3), E370(8), S375(3), F382(2), N425(8), M426(4), W427(9), V430(6), G473(12), M475(7), D474(5), I371(3)	1...W427 (T-shaped-interaction)

Notes: ^1^ Donors of the hydrogen bonds relating to the ligands are shown first, followed by the corresponding acceptors of the gp120 residues. The residues of gp120 are in brackets in one-letter code. Subscripts of nitrogen and hydrogen atoms match their numbering in Figure 4. ^2^ The gp120 residues forming van der Waals contacts with the identified compounds are given in one letter code. The number of the contacts is shown in round brackets. ^3^ The functional groups of ligands and numbers of their aromatic rings (Figure 4) are shown first for salt bridges and-interactions, respectively.

**Table 3 viruses-11-00746-t003:** Values of binding free energy (∆G) and K_d_ calculated for the PM7-based complexes of the identified compounds and NBD-11021 with gp120.

Compound	I	II	III	IV	V	NBD-11021
∆G, kcal/mol ^1^	−10.6	−9.8	−9.9	−9.5	−9.8	−7.8
K_d_ (µM) ^2^	0.0075	0.2767	0.4675	0.5086	0.0593	2.1
∆G, kcal/mol ^3^	−11.1	−9.0	−8.7	−8.6	−9.9	−8.0

Notes: ^1^ The ∆G values according to the QuickVina 2 scoring function; ^2^ The values of K_d_ calculated using a neural-network-based scoring function NNScore 2.0; ^3^ The ∆G values estimated from those of K_d_ by the formula ∆G = R × T × ln(K_d_) (where ∆G is the binding free energy, R is the universal gas constant, T is the absolute temperature equal to 310 K) [86].

**Table 4 viruses-11-00746-t004:** Mean values of binding free energy <ΔG> for the complexes of the CD4-mimetic candidates and NBD-11021 with gp120 and their standard deviations ΔG_STD_
^1^.

Compound	<ΔH> kcal/mol	(ΔH)_STD_ kcal/mol	<TΔS> kcal/mol	(TΔS)_STD_ kcal/mol	<ΔG> kcal/mol	ΔG_STD_ kcal/mol
I	−49.40	5.06	−20.89	7.01	−28.51	7.65
II	−32.64	3.95	−18.61	6.28	−14.03	7.42
III	−34.30	3.03	−23.69	9.72	−10.61	7.18
IV	−29.00	4.51	−19.50	8.12	−9.50	6.29
V	−27.93	4.02	−20.35	9.06	−7.58	5.91
NBD-11021	−30.41	3.60	−22.42	9.89	−7.99	7.52

Note: ^1^ <ΔH> and <TΔS> are the mean values of enthalpic and entropic components of free energy respectively; (ΔH)_STD_ and (TΔS)_STD_ are standard deviations corresponding to these values.

**Table 5 viruses-11-00746-t005:** Averages of the binding enthalpy for the amino-acid residues of gp120 bound to the CD4-mimetic candidates ^1^.

Residue of gp120	CD4-Mimetic Candidate				
	I	II	III	IV	V
	Residue Contribution to the Binding Enthalpy (kcal/mol) ^2, 3^				
Gly-128	−0.99	-	-	-	-
Ala-129	−0.78	-	-	-	
Gly-194	−0.59	-	-	-	-
Val-255	−1.23	−1.32	−1.35	−1.08	−1.04
Ser-256	−0.68	−1.03	-	-	−0.80
Thr-257	−1.14	−1.65	-	−1.39	−1.29
Asp-368	−0.92	−1.10	−0.15	−0.85	−1.42
**Glu-370**	**−2.00**	**−2.63**	**−1.70**	**−3.10**	**-**
Ile-371	-	−0.82	-	-	−0.91
Ser-375	−1.76	−2.36	-	−0.82	−0.98
Phe-376	−0.66	−0.59	-	-	-
Phe-382	−0.78	−0.69	−0.68	−0.65	-
Tyr-384	−0.75	−0.64	-	-	-0.68
Ile-424	−0.76	−0.74	−1.09	−0.88	−0.59
**Asn-425**	**−6.49**	**−3.48**	**−2.68**	**−2.96**	**−2.76**
**Met-426**	**−1.70**	**−1.63**	**−2.17**	**−3.00**	**−1.57**
**Trp-427**	**−3.89**	**−2.73**	**−4.49**	**−5.1**	**−3.18**
Gln-428	−0.68	-	−0.99	−0.88	-
Lys-429	−1.85	-	−1.96	−1.78	-
Val-430	−3.63	-	−1.50	−2.10	−0.72
Gly-431	−1.10	-	-	-	-
Gly-472	-	-1.01	-	-	-0.54
**Gly-473**	**−1.54**	**−1.54**	**−1.29**	**−1.24**	**−2.83**
**Asp-474**	**−3.07**	**−1.21**	**−2.14**	**−1.56**	**−2.71**
**Met-475**	**−3.60**	**−1.93**	**−3.39**	**−1.65**	**−2.55**
Arg-476	-	-	−0.54	-	-

Notes: ^1^ The MM/GBSA.py procedure of AmberTools 11 [73] was used to decompose the enthalpic component of the binding free energy into the contributions from each amino acid of gp120. ^2^ Data for the gp120 residues with the binding enthalpy ≤−0.5 kcal/mol are presented. ^3^ The gp120 residues that greatly contribute to the enthalpic component of binding free energy are highlighted by bold.

**Table 6 viruses-11-00746-t006:** Intermolecular hydrogen bonds in the dynamic structures of the identified compounds bound to gp120.

Ligand	Hydrogen Bond ^1^
I	O1...HG2[Glu370; 71,3%], O2...HG2[Glu370; 20,9%], O1...HB3[Asn425; 15,6%], N4...HD21[Asn425; 99,8%], O2...HD22[Asn425; 39,7% ], N1...HB2[Trp427; 51,9%], O1...HB3[Trp427; 36,6%], N4...HA[Val430; 83,5%], N4...H[Gly431; 85,0%], N1...HA[Asp474; 35,0%], N2...HA[Asp474; 77,3%], N3...HA[Trp427; 71,1%], N1...H[Met475; 78,3%]; N2...H[Met475; 94,7%], N1...HB2[Met475; 52,0%], N2...HB2[Met475; 16,6%], N1...HG2[Met475; 32,9%], H1...O[Val255; 49,4%], H1...O[Ser375; 37,4%], H1... O_D2_[Asp368; 15,3%], H2...N[Phe376; 19,1%], H3...OD2[Asp474; 20,0%], H4...O[Gly473; 26,3%], H5...O[Trp427; 71,5%], H6...O[Trp427; 71,3%], H7...ND2[Asn425; 49,3%], H7...O[Met426; 71,5%], H8...O[Gly128; 39,0%], H9...O[Gly128; 42,4%], H10...O[Gly194; 22,3%]
II	N2...HG2[Glu370; 26,0%], N1...HG2[Glu370; 34,7%], N3...HA[Asp474; 26,2%], H1...O[Val255; 27,5%], H1... O_D1_[Asp368; 17,1%], H2...O[Phe376; 42,7%], H3...OH[Tyr384; 26,3%], H4...O[Asn425; 47,2%], H5...OD1[Asp368; 1,3%]
III	N2...HB3[Asn425; 44,2%], N3...HB3[Asn425; 25,5%], N5...HA[Trp427; 42,7%], N6...HA[Trp427; 50,2%], N4...HB2[Trp427; 39,5%], N1...HB3[Trp427; 31,4], N2...HB3[Trp427; 39,4%], N4...HA[Asp474; 82,8%], N5...HA[Asp474; 27,2%], N4...H[Met475; 84,6%], N5...H[Met475; 32,6%], N4...HB2[Met475; 30,7%], H1... O_D2_[Asp368; 18,7%], H1...O[Trp427; 68,3%], H2...O[Trp427; 45,3%]
IV	N1...HB3[Asn425; 26,8%], N2...HB3[Asn425; 53,5%], N3...HA[Val430; 44,2%], N3...H[Gly431; 40,9%], O1...HA2[Gly473; 45,4%], N4...HA3[Gly473; 36,2%], H4...O[Met426; 95,8%], H5...O[Gly473; 52,7%], H1...O[Asn425; 62,4%], H1... O_D2_[Asp368; 12,5%], H2...SD[Met475; 26,1%], H3...O[Gly473; 50,1%], H7...OD2[Asp368; 16.3%], H7...OD1[Asp368; 12.1%], H6...OD2[Asp368; 7.8%], H6…OD1[Asp368; 5.4%]
V	N2...HB2[Trp427; 29,5%], N1...HB2[Trp427; 27,5%], N2...HA[Asp474; 47,2%], N2...H[Met475; 48,8%], N1...HB2[Met475; 26,9%], H1...O[Ser256; 30,5%], H1... O_D2_[Asp368; 15,3%], H2...O[Asn425; 73,6%], H3...O[Gly473; 44,8%]

Note: ^1^ Donors and acceptors of the hydrogen bonds relating to the ligands are shown first, followed by the corresponding functional groups of the gp120 amino acids. Subscripts of the ligand oxygen, nitrogen and hydrogen atoms match their numbering in Figure 3. The residues of gp120 and percentage occupancies of hydrogen bonds are indicated in square brackets.

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
