# Peer review of "In Silico Identification of Novel Aromatic Compounds as Potential HIV-1 Entry Inhibitors Mimicking Cellular Receptor CD4"

_viruses, 2019, doi:10.3390/v11080746_

Round 1
Reviewer 1 Report
The manuscript by Andrianov and co-workers utilize in silico approach to identify mimemic compounds that can inhibit critical interactions of the HIV-1 envelope gp120 protein with host cell receptor CD4. The studies have demonstrated that they have identified five compounds that have the potential for further development into novel HIV-1 entry inhibitors.
The manuscript needs minor revision, the introduction is very long and should be shortened.
Author Response
Reviewer 1
Comment. The manuscript needs minor revision, the introduction is very long and should be shortened.
Response: We thank the reviewer for the suggestion. We have shortened the Introduction accordingly (please see the revised version with changes tracked).
Reviewer 2 Report
The authors performed an in silico screen to identify novel drug-like inhibitors that block gp120-CD4 binding, using a panel of cutting-edge computational approaches that involved molecular docking to a gp120 monomer followed by semiempirical quantum chemical calculations and assessment of the binding affinity of identified ligands to gp120 by molecular dynamic simulations and free energy calculations. Five 5 best compounds from the screen were selected for more detailed characterization of their binding modes to gp120, which is predicted to be primarily mediated by multiple van der Waals interactions with the gp120 residues and the hydrogen bond with Asp-368 of gp120. Encouragingly, the binding affinities and free energies of these five compounds are predicted to be better than the lead CD4 mimetic compound NBD-11021.
The reported findings are interesting and the top candidates appear promising. However, having these compounds synthesized and tested for antiviral activity would greatly increase the value of computational efforts which are generally known to have somewhat poor predictive power.
A major concern is related to the choice the structure a monomeric gp120 in complex with soluble CD4 and 17b antibody fragment which is a surrogate for the Env-CD4-coreceptor complex. First, monomeric gp120 is free of quaternary restrictions of a trimer and thus may not be an ideal model of a native Env. It is also surprising that the authors choose to model compound binding to a CD4-bound conformation, as opposed to the native unliganded structure which should ideally be targeted by drugs.
Author Response
Reviewer 2
Comment 1. The reported findings are interesting and the top candidates appear promising. However, having these compounds synthesized and tested for antiviral activity would greatly increase the value of computational efforts which are generally known to have somewhat poor predictive power.
Response. We thank the reviewer for the constructive comments, which have inspired us to strengthen the manuscript. Indeed, our original aims are to identify the lead compounds through virtual screening, and then synthesize these compounds and test their anti-HIV-1 activity, when we applied for the Belarusian Republican Foundation for Fundamental Research. We did get the grant, but the budget that we obtained is very limited, which is enough for virtual screening only, not for chemical synthesis of the compounds. Therefore, we will have to apply for a new grant to support the chemical synthesis. Considering that this application process will take one or two years, we decided to publish the virtual screening results, which may attract some researchers who have the funds and are interested in synthesis of these compounds.
Comment 2. A major concern is related to the choice the structure a monomeric gp120 in complex with soluble CD4 and 17b antibody fragment which is a surrogate for the Env-CD4-coreceptor complex. First, monomeric gp120 is free of quaternary restrictions of a trimer and thus may not be an ideal model of a native Env. It is also surprising that the authors choose to model compound binding to a CD4-bound conformation, as opposed to the native unliganded structure which should ideally be targeted by drugs.
Response. We are grateful for the reviewer's insightful comments. We agree that the use of the available structures of the monomeric gp120 core may limit their applicability as appropriate targets for virtual screening of novel CD4 mimetics candidates that bind to the HIV-1 gp120 subunit. However, the computational evaluation of different HIV-1 gp120 conformations as targets for de novo docking of first- and second-generation small-molecule CD4 mimics showed that in silico neutralization of the highly conserved residue Asp-368gp120 is critically necessary to obtain the correct orientation of small-molecule CD4 mimetics in their binding site when docking against the monomeric gp120 core, which correlates with IC50's measured in CD4 binding competition ELISA and with KD's measured on gp120 core monomer (Moraca et al. J. Chem. Inf. Model. 2016, 56, 2069-2079). As shown in our study, all the identified compounds form hydrogen bond and van der Waals contacts with Asp-368gp120, suggesting that the data obtained from the analysis of the docked ligand/monomeric gp120 complexes provide correct information on the interaction modes and binding affinity profiles of these molecules and gp120. This assumption is supported by the optimization of these complexes using the semiempirical quantum chemical method PM7 which improves the accuracy of docking models based on classical force field, as evidence from the study of Sulimov & coauthors (Adv. Bioinformatics 2017, 5, 1-6). Furthermore, the findings of molecular dynamics of the analyzed docked structures that represents the flexibility of both ligands and a target protein also validate the assumption made. Finally, there are some active anti-HIV compounds that were obtained by virtual screening using the X-ray crystal structure of the CD4-bound HIV-1 gp120 core and have provided lead scaffolds for successful development of novel potent HIV-1 entry inhibitors (e.g., LaLonde et al. Bioorg. Med. Chem. 2011, 19, 91–101; Courter et al. Acc. Chem. Res. 2014, 47, 1228–1237), indicating the validity of application of this structure to screen new CD4 mimetics candidates targeting CD4-binding site of gp120. Nevertheless, the study of Moraca & coauthors supports the hypothesis that the 4NCO trimeric gp120 structure represents more appropriate target for optimization of CD4-mimetic candidates with advantages over both the 4TVP trimer and gp120 core monomer (Moraca et al. J. Chem. Inf. Model.2016. 56, 2069-2079). Therefore, the further advancement of our study suggests the use of the 4NCO trimeric gp120 structure as a target for optimization of the lead compound to arrive at CD4 mimetics with improved anti-HIV potency and pharmacokinetic properties.
The justification for using in our work the X-ray structure of the CD4-bound HIV-1 gp120 core as the target for virtual screening potential CD4 mimetics candidates is given in the last paragraph of the Results and Discussion section along with appropriate reference citations (please see the revised manuscript with changes tracked).
Round 2
Reviewer 2 Report
No further concerns beyond those raised in the first revision.
Author Response
Reviewer 2
Comment 1. No further concerns beyond those raised in the first revision.
Response. We are very grateful to you for valuable and absolutely right comments concerning selection of the X-ray CD4/gp120/Fab17b structure as a target for prediction of the interaction modes and binding affinity profiles of the designed compounds and gp120. We added to the previous manuscript version a number of additional data indicating the validity of application of this structure to screen new CD4 mimetics candidates targeting CD4-binding site of gp120 (along with appropriate reference citations).
Certainly, the 4NCO trimeric gp120 structure is more appropriate target for structure-based design of the gp120-directed inhibitors compared with the 4TVP trimer and gp120 core monomer However, it takes huge computational resources to perform quantum chemical calculations and molecular dynamics simulations for a large number of CD4 mimetics candidates. Nevertheless, the further advancement of this study proposes the application of the 4NCO trimeric gp120 structure at the stage of the optimization of the lead compound to arrive at CD4 mimetics with improved anti-HIV potency and pharmacokinetic properties (this is also given in a new version of the manuscript).
All the corrections made in the current paper version are highlighted by yellow.
We are very thankful to you again for your interest in this work and constructive and thoughtful comments, which prompted us to improve the manuscript.